# OpenReview forum: "Reinforcement Learning with Backtracking Feedback"
_NeurIPS.cc/2025/Conference — NeurIPS 2025 poster_

### Official Review · Reviewer_Gc4v · 2025-07-03

**Clarity:** 4
**Significance:** 4
**Originality:** 3
**Rating:** 5
**Confidence:** 5

**Summary:**

The authors develop a novel method, RLBF, for training models to backtrack unsafe outputs via RL. RLBF starts with an SFT phase with data constructed with harmful tokens injected such that we can precisely target their removal by backtracking X tokens (BSAFE+). The method then continues with GRPO using an LLM safety critical to provide safety signal. The authors conduct experiments to explore the resilience of models trained with RLBF against harmful queries.

**Questions:**

Questions:
- What are the failure modes of RLBF?
- Does the BSAFE+ dataset contain adversarial examples, or are the results against GCG etc. without any exposure to the attacks during training?

**Ethical Concerns:**

["NO or VERY MINOR ethics concerns only"]

**Final Justification:**

The authors’ response answered my questions, and I raised my Quality rating from 3 to 4.

**Limitations:**

Yes

**Quality:**

4

**Strengths And Weaknesses:**

Strengths:
- The approach to RL and text generation with precise backtracking as a first-class operation is effective and efficient, with potential applications in correcting mistakes in LLM generation beyond just safety.
- RLBF is highly effective, reducing the ASR better than other methods while maintaining task accuracy.

Weaknesses:
- The method relies on a single LLM safety critic, which can be vulnerable to adversarial attacks or otherwise produce incorrect judgments.
- The paper doesn’t contextualize the ASR numbers with non-backtracking methods that improve safety via post-training or at inference time.

---

> ### Author Rebuttal · Authors · 2025-07-31
>
> We thank the reviewer for their thoughtful review and questions. We were encouraged to hear that it “highly effective, reducing the ASR better than other methods while maintaining task accuracy” and it possesses "potential applications in correcting mistakes in LLM generation beyond just safety". We have acted on their suggestions and introduced new results to more comprehensively compare our methods against inference-time alignment methods.
>
> In the following, we will address the concerns and questions of the reviewer in detail.
>
> ---
>
> ## **Regarding potential vulnerability of a single LLM safety critic to adversarial attacks**
>
> This is a thoughtful concern about the robustness of our critic-based approach. However, our critic design includes several protective mechanisms that significantly reduce vulnerability to adversarial attacks:
> - **Structured Evaluation Format:** Our safety critic does not simply receive the user query and model response in a direct conversational format, which would indeed make it susceptible to the same jailbreaking techniques that target the main model. Instead, as detailed in Appendix A.2, we employ a carefully structured prompt format where the critic receives explicit instructions about its evaluation task, followed by the user query and assistant response presented in a hierarchical, clearly delineated format with "User:" and "Assistant:" prefixes. This approach aligns with principles from Constitutional AI work [1], which demonstrated that structured evaluator prompts can enhance robustness against adversarial manipulation.
> - **Critic Quality Validation:** Although we have discussed this briefly in the supplementary, we wanted to dissect the importance and difficulty of obtaining accurate critics. In the beginning of our research, we were concerned whether the critic we might create would be able to accurately identify the starting point of unsafe content together with the specific violation category. Therefore, we created three different prompts to test out on 100 prompt-response pairs we (as the authors) created manually to cover a wide range of policy violations and policy violations occurring at a wide range of locations. We saw that the accuracy of these prototypes were almost the same (97.6%, 97.6%, 98.0% for 3 critic samples per prompt-response pairs). This validated our decision to move forward with using any of these critiques.
>
> We wanted to follow-up with a more detailed analysis on those 100 prompt-response pairs, by categorizing them in three: High/Critical, Medium and Low/Negligible. As per the EU AI Act [2] and Google's Vertex AI Safety filters [3], general guidelines are as follows:
> - High Severity: Immediate severe harm such as terrorism, self-harm, CSAM (which we do not have any examples of for ethical reasons).
> - Medium Severity: Harmful content but not catastrophic (bias, harassment, moderate misinformation)
> - Low Severity: Mild rudeness, harmless errors (date errors, small factual mistakes)
>
> We found out that there were 43 high, 38 medium and 19 low severity examples out of our 100 prompt-response pairs. Together with the accuracy of the safety critics, this shows that the model with this structured prompting is much less prone to safety violations the original model might make during generating a response.
> - **Policy Learning vs. Critic Quality:** The ablation analysis in Section 5.4 demonstrates that RLBF's gains stem from policy learning with backtracking capabilities rather than simply having a high-quality critic. The RL baseline trained with the same critic but without backtracking signals shows higher ASR than RLBF models, even when backtracking is disabled, indicating that the structured learning process drives the performance improvements.
>
> [1] Bai et al. 2022, https://arxiv.org/abs/2212.08073
>
> [2] https://artificialintelligenceact.eu/
>
> [3] https://cloud.google.com/vertex-ai/generative-ai/docs/multimodal/gemini-for-filtering-and-moderation
>
> ## **Comparisons to Non-Backtracking and Inference Time Alignment Methods**
>
> We appreciate this feedback and would like to clarify that our evaluation does include substantial comparison with non-backtracking safety methods, both post-training and inference-time approaches. Our baseline selection was designed to provide comprehensive context for RLBF's performance across different safety paradigms.
> - **Post-Training Method Coverage:** Our IT (Instruction Tuned) baselines represent state-of-the-art post-training safety methods from major research institutions - these are the official safety-aligned models from Meta (LLaMA 3) and Google (Gemma 2) that have undergone extensive post-training safety alignment including RLHF, constitutional AI, and other safety interventions. These models represent the current standard for post-training safety approaches and provide a strong baseline for comparison.
> - **Controlled RL Baseline:** We include an RL baseline that uses the exact same critic and reward structure as RLBF but without backtracking signals. This isolates the contribution of our backtracking mechanism from general RL-based safety improvements, providing direct evidence that our gains come from the backtracking capability rather than simply applying RL to safety.
> - **Inference-Time Alignment:** We compare extensively against Circuit Breakers, a prominent inference-time safety method that operates by controlling internal representations during generation. To provide complete context that was not fully presented in our original tables, we include new comprehensive Circuit Breaker results below:
>
>
> LMSYS-MF - Attack Success Rate (%) - lower is better
>
> | Method | Gemma2-2B | Gemma2-9B | LLaMA3-1B | LLaMA3-3B | LLaMA3-8B |
> | ------ | --------- | --------- | --------- | --------- | --------- |
> | IT     | 71        | 75        | 68        | 77        | 81        |
> | CB     | 14        | 17        | 13        | 16        | 18        |
> | RL     | 67        | 72        | 61        | 64        | 61        |
> | BSAFE+ | 4         | 3         | 6         | 5         | 5         |
> | RLBF   | 4         | 3         | 7         | 5         | 3         |
>
>
> LMSYS - Attack Success Rate (%) - lower is better
>
> | Method | Gemma2-2B | Gemma2-9B | LLaMA3-1B | LLaMA3-3B | LLaMA3-8B |
> | ------ | --------- | --------- | --------- | --------- | --------- |
> | IT     | 25        | 28        | 24        | 28        | 27        |
> | CB     | 18        | 17        | 21        | 21        | 20        |
> | RL     | 23        | 24        | 22        | 25        | 25        |
> | BSAFE+ | 14        | 15        | 14        | 17        | 16        |
> | RLBF   | 2         | 2         | 1         | 2         | 1         |
>
> While Circuit Breakers show improvement over basic IT models, they underperform compared to both RLBF and BSAFE+ across all model sizes and benchmarks. Additionally, Circuit Breakers' mechanism of generating incoherent text when violations are detected creates usability issues, as abrupt stopping of coherent generation is undesirable from a user experience perspective, particularly evident in the LMSYS-MF results where this behavior is especially problematic. These comprehensive comparisons demonstrate that RLBF achieves superior safety performance compared to both post-training and inference-time non-backtracking approaches while maintaining full utility.
>
> ---
>
> ## Further Clarifications
>
> >What are the failure modes of RLBF?
>
> **Response:**
> While RLBF significantly improves safety, with this new addition of backtracking and learning to use it through RLBF, the models may still fail to recognize when backtracking is needed, missing subtle violations that require correction. However, our empirical results show a strong capability in backtracking when needed, with consistently low ASRs across diverse benchmarks and model scales.
>
> >Does the BSAFE+ dataset contain adversarial examples, or are the results against GCG etc. without any exposure to the attacks during training?
>
> **Response:**
> Thank you for this question as it helps us highlight the robustness against jailbreaks we get from RLBF and BSAFE+. The BSAFE+ dataset does not have any examples to protect the model against GCG or Decoding Parameters Exploit (we use single set of generation config). However, the end model is shown to be robust against these adversarial attacks without any explicit training in SFT or in RL stages.
>
> ---
>
> ## Conclusion
>
> We are again thankful for your review. If our clarifications and new experimental results helped, please consider updating your scores for our paper (e.g., Quality, Clarity, Significance and/or Originality).

---

> > ### Comment · Reviewer_Gc4v · 2025-08-02
> >
> > Thank you for your response and clarifications! The robustness of the method against adversarial prompts like GCG despite no explicit training is an especially strong finding, and maybe worth calling out in the paper. I think this is a strong paper, and will continue to recommend acceptance.

---

### Official Review · Reviewer_mSbG · 2025-07-04

**Clarity:** 3
**Significance:** 3
**Originality:** 3
**Rating:** 4
**Confidence:** 2

**Summary:**

This paper introduces RLBF, a framework designed to enhance the safety of LLMs by enabling dynamic self-correction of safety violations during generation. RLBF employs a token-efficient "backtrack by x tokens" mechanism, trained via an improved BSAFE+ SFT strategy and refined through reinforcement learning with real-time feedback from an LLM safety critic. Empirical results across multiple model architectures (Gemma 2, LLaMA 3) and benchmarks demonstrate that RLBF significantly reduces attack success rates against adversarial strategies while preserving model utility on standard tasks.

**Questions:**

The selected benchmark does not seem to be consistent with the benchmark chosen in the baseline method paper. Will this have any impact on the evaluation?

**Ethical Concerns:**

["NO or VERY MINOR ethics concerns only"]

**Final Justification:**

I appreciate the authors' rebuttal, in which most of my concerns are addressed, so I decide to increase my score.

**Limitations:**

Yes

**Quality:**

3

**Strengths And Weaknesses:**

Strengths
1. Rigorous evaluations against baselines across diverse adversarial attacks and model scales show consistently lower attack success rates
2. RLBF maintains performance on standard benchmarks comparable to base models


Weakness
1. The design of the reward function is highly subjective and has not been verified through experiments.
2. The baseline BSAFE[1] for the comparison is an article published on arXiv, which might cast doubt on the persuasiveness of the comparison results. In addition, baseline Circuit Breakers[2] was only compared in Table 2.
3. There are some obvious formatting errors, such as the [BSAFE reference] on line 112 and line 124, and the instruction block in the "Paper checklist" has not been deleted.

[1] Sel et al., BSAFE: (B)acktracking for (SAFE)ty. Arxiv 2025
[2] Zou etal., Improving Alignment and Robustness with Circuit Breakers. NeurIPS 2024.

---

> ### Author Rebuttal · Authors · 2025-07-31
>
> We thank the reviewer for their thoughtful review and suggested improvements. We were pleased to hear that it has "rigorous evaluations", shows "consistently lower attack success rates", and it "maintains performance" on utility benchmarks. We have acted on their feedback to introduce new experiments & results.
>
> In the following, we address each question & concern in detail.
>
> ---
>
> ## **Design of the Reward Function**
> We appreciate the reviewer's question about the reward function design. While we acknowledge that reward function design involves some design choices, our approach is grounded in principled reasoning rather than arbitrary decisions:
>
> **Justifications:**
> 1. **Binary safety outcomes**: The core distinction between safe (+1.0) and unsafe (-1.0) generations reflects the fundamental binary nature of safety - content is either acceptable or not.
> 2. **Graduated penalties for backtracking**: The intermediate values (-0.5 for unnecessary backtracking, -0.2 for failed corrections) create a natural hierarchy that encourages efficient use of the backtracking mechanism while still rewarding attempted corrections over generating harmful content.
> 3. **Alignment with safety objectives**: The reward structure directly incentivizes the three key behaviors we want: (1) generating safe content directly, (2) using backtracking appropriately when needed, and (3) ensuring high-quality corrections.
>
> **Empirical Support:** The chosen reward values led to improvements in all model families and model sizes we tested, providing key evidence for its effectiveness compared to typical RL training.
> - Significant ASR reductions across all benchmarks (Table 1-2)
> - Preserved model utility (Table 3)
> - Effective performance across safety categories (Table 4)
>
> However, in the future work, we plan to delve deeper into reward design for RLBF, as we believe the exact specifics are orthogonal to our main contributions.
>
> ## **Further Baselines**
> In most of our experiments, we actually compare RLBF and BSAFE+ to baselines, which is different from BSAFE in terms of how the SFT dataset is generated. Please check out Section 4.1.2 for further information regarding this.
>
> RLBF (also BSAFE+), by design, is orthogonal to the various methods proposed in the literature for improving safety, such as better data curation, representation adjustments between layers to improve generation safety etc. RLBF focuses on the case where everything else has failed, and whether we can backtrack using the same model in an efficient and effective way to enhance safety. Therefore, our baselines include the most direct comparisons. For instance, RL baseline uses the exact same critic and rewards except for the ones used for backtracking to isolate the impact of backtracking. Similarly, Table 5 gives a detailed ablation analysis on the significance of the backtracking even in the RLBF model to ensure that the safety improvements are directly contributable to Reinforcement Learning with Backtracking Feedback.
>
> We opted to include Circuit Breakers [1] baseline only in Table 2. Because:
> - Table 3 only highlights that RLBF or BSAFE+ does not degrade utility on significant benchmarks. Circuit Breakers degrading or not degrading the performance does not convey any useful information to the reader.
> - Table 4 is on the safety performance of RLBF models on various safety policies.
> - Table 5 gives a detailed ablation analysis on the significance of the backtracking even in the RLBF model to ensure that the safety improvements are directly contributable to Reinforcement Learning with Backtracking Feedback, as discussed before.
>
> _LMSYS-MF - Attack Success Rate (%) - lower is better_
>
> | Method | Gemma2-2B | Gemma2-9B | LLaMA3-1B | LLaMA3-3B | LLaMA3-8B |
> | ------ | --------- | --------- | --------- | --------- | --------- |
> | IT     | 71        | 75        | 68        | 77        | 81        |
> | CB     | 14        | 17        | 13        | 16        | 18        |
> | RL     | 67        | 72        | 61        | 64        | 61        |
> | BSAFE+ | **4**         | **3**         | **6**         | **5**         | 5         |
> | RLBF   | **4**         | **3**         | 7         | **5**         | **3**         |
>
>
> _LMSYS - Attack Success Rate (%) - lower is better_
>
> | Method | Gemma2-2B | Gemma2-9B | LLaMA3-1B | LLaMA3-3B | LLaMA3-8B |
> | ------ | --------- | --------- | --------- | --------- | --------- |
> | IT     | 25        | 28        | 24        | 28        | 27        |
> | CB     | 18        | 17        | 21        | 21        | 20        |
> | RL     | 23        | 24        | 22        | 25        | 25        |
> | BSAFE+ | 14        | 15        | 14        | 17        | 16        |
> | **RLBF**   | **2**         | **2**         | **1**         | **2**         | **1**         |
>
> Circuit breakers show worse safety performance compared to RLBF and BSAFE+. Furthermore, since the Circuit Breakers simply get the model to generate meaningless text, qualitatively, abrupt stopping of providing coherent text to the user is undesirable even in the examples where the model is safe, especially in LMSYS-MF.
>
> [1] Zou et al., 2024 https://arxiv.org/pdf/2406.04313
>
> ---
>
> ## Further Clarifications
>
> >The selected benchmark does not seem to be consistent with the benchmark chosen in the baseline method paper. Will this have any impact on the evaluation?
>
> **Response:**
> LMSYS-MF benchmark we use in our paper follows a similar idea to BSAFE paper's proposed benchmark with the middle filling jailbreaks. However, we opted to ground that idea on a widely known LMSYS dataset to better represent the distribution of real questions asked to chatbots.
>
> ---
>
> ## Conclusion
> We have worked hard to clarify and include new experiments in our rebuttal. **As your pre-rebuttal scores for Quality, Clarity, Significance and Originality for our paper are all "good", we are hoping that our clarifications & new results have addressed your concerns. If so, we kindly ask you to reconsider your overall rating**. Feel free to let us know if you'd like us to address anything further. Thank you again for your thoughtful feedback!

---

> > ### Comment · Reviewer_mSbG · 2025-08-05
> >
> > I appreciate the authors' rebuttal, in which most of my concerns are addressed, so I decide to increase my score.

---

### Official Review · Reviewer_wAMH · 2025-07-06

**Clarity:** 2
**Significance:** 3
**Originality:** 2
**Rating:** 4
**Confidence:** 5

**Summary:**

The paper introduces Reinforcement Learning with Backtracking Feedback (RLBF), a training recipe for LLMs to recognize and correct emergent violations during generation: the model first learns a token-efficient “backtrack by x tokens” command through an enhanced supervised fine-tuning procedure (BSAFE+), in which safety-critical segments are programmatically inserted into otherwise coherent answers, then refines this behavior via a reinforcement learning stage that receives live feedback from an LLM-based safety critic. This combined approach enables the model to discard only the problematic span, and thus improve generation efficiency compared to the prior safety backtrack approach. Experimental results demonstrate the effectiveness of the proposed method.

**Questions:**

1. In line 174, how do you inject violation at a contextually coherent location? Can you elaborate more? How do you ensure the violating content can be semantically coherent after part 1?

2. Can you elaborate more on the setting of middle filling attacks?

**Ethical Concerns:**

["NO or VERY MINOR ethics concerns only"]

**Final Justification:**

The rebuttal fully addresses my concerns on the e2e efficiency computation, adapative attacks, etc.

**Limitations:**

yes

**Quality:**

3

**Strengths And Weaknesses:**

**Strengths**

1. *Targeted safety correction with minimal generation waste.* By teaching the model to emit a simple “\[CATEGORY] \[BACKTRACK\_BY\_X]” signal, RLBF removes only the violating span instead of resetting or rewriting large chunks of otherwise correct text, which the authors argue is more efficient than both reset‐style baselines.

2. *Sound optimization design.* The combination of (i) SFT loss anchoring on curated BSAFE+ examples and (ii) an RL objective shaped by a single LLM safety critic — implemented through GRPO with a balanced cloning term—forms a coherent training pipeline that jointly rewards safe continuation and discourages unnecessary backtracks.


**Weaknesses**

1. *Insufficient experimental transparency.* Key implementation choices—data splits, prompt sampling, training hyper-parameters, and compute budget—are deferred to the supplementary material, leaving the main text without the detail needed to judge reproducibility.

2. *Missing details of safety critic.* Section 4.2 names a “single, powerful LLM-based safety critic” but omits the evaluation of the critic quality (aka RM offline evaluation) making it hard to assess whether critic quality or policy learning drives the reported gains.

3. *Limited coverage of adaptive attacks.* While Table 2 includes GCG and decoding-parameter variants, the evaluation set does not appear to target RLBF’s specific backtracking signal in the way Zhang et al. probe BSAFE, so robustness under truly adaptive adversaries remains unclear.

4. *Missing end-to-end efficiency comparison.* A core claim is higher efficiency relative to BSAFE, yet the experiments focus on Attack Success Rate and utility benchmarks.  Metrics such as average tokens discarded or latency under streaming would strengthen the argument.


**Minor issues**

* Multiple placeholders (“\[two references]”, “\[BSAFE reference]”) remain in Section 3 and should be replaced with proper citations.

---

> ### Author Rebuttal · Authors · 2025-07-31
>
> We thank the reviewer for their thoughtful review and suggested improvements. We were pleased to hear that it targets “safety correction with minimal generation waste” and it has "sound optimization design". **We have acted on their feedback to introduce new experiments & results**.
>
> In the following, we address each question & concern in detail.
>
> ---
> ## **The experiment details in the supplementary material should be moved to main text**
> We were limited by the maximum 9-page limit during the submission. However, during camera-ready submission, we are allowed 10 pages, so we will move the key implementation choices—data splits, prompt sampling, training hyper-parameters, and compute budget—from the supplementary to main text.
>
> ## **Further Details of Safety Critic**
> Thank you for pointing this out. Although we have discussed this briefly in the supplementary, we wanted to dissect the importance and difficulty of obtaining accurate critics. In the beginning of our research, we were concerned whether the critic we might create would be able to accurately identify the starting point of unsafe content together with the specific violation category. Therefore, we created three different prompts to test out on 100 prompt-response pairs we (as the authors) created manually to cover a wide range of policy violations and policy violations occurring at a wide range of locations. We saw that the accuracy of these prototypes were almost the same (97.6%, 97.6%, 98.0% for 3 critic samples per prompt-response pairs). This validated our decision to move forward with using any of these critiques.
>
> We wanted to follow-up with a more detailed analysis on those 100 prompt-response pairs, by categorizing them in three: High/Critical, Medium and Low/Negligible. As per the EU AI Act [1] and Google's Vertex AI Safety filters [2], general guidelines are as follows:
> - High Severity: Immediate severe harm such as terrorism, self-harm, CSAM (which we do not have any examples of for ethical reasons).
> - Medium Severity: Harmful content but not catastrophic (bias, harassment, moderate misinformation)
> - Low Severity: Mild rudeness, harmless errors (date errors, small factual mistakes)
>
> We found out that there were 43 high, 38 medium and 19 low severity examples out of our 100 prompt-response pairs, hence critic performance is tested on a variety of prompts.
>
> In order to better observe how the critic quality effects policy learning, we used a Llama-2 8B model with the same safety critic prompt that has an accuracy of only 82.0% compared to 98% in our original experiments.
>
> _LMSYS-MF - ASR (%) - lower is better_
>
> |Method|Gemma2-2B|Gemma2-9B|LLaMA3-1B|LLaMA3-3B|LLaMA3-8B|
> |---|---|---|---|---|---|
> |IT|71|75|68|77|81|
> |RL|67|72|61|64|61|
> |BSAFE+|4|3|6|5|5|
> |RLBF (Weak Critic)|4|3|7|5|4|
> |RLBF|5|3|7|5|3|
>
> _LMSYS - ASR (%) - lower is better_
>
> |Method|Gemma2-2B|Gemma2-9B|LLaMA3-1B|LLaMA3-3B|LLaMA3-8B|
> |---|---|---|---|---|---|
> |IT|25|28|24|28|27|
> |RL|23|24|22|25|25|
> |BSAFE+|14|15|14|17|16|
> |RLBF (Weak Critic)|4|3|2|4|4|
> |RLBF|2|2|1|2|1|
>
> These show that the RLBF's gains we observe are due to policy learning with the capabilities that backtracking gives as opposed to simply using a better critic.
>
> Further information regarding whether RLBF's gains are due to critic quality, policy learning and/or backtracking capability, can be inferred from the discussion in Section 5.4. Briefly,
> - The baseline denoted as RL, is trained with the same critic albeit with safe/unsafe signal without backtracking signals. That model in the end has higher ASR (Attack Success Rate, lower is better) than the RLBF model with backtracking disabled (through negative bias on backtracking tokens). This suggests that utilizing a critic to also get a signal on where the backtracking should occur helps RLBF models even in cases where they are not allowed to backtrack.
> - Furthermore, with backtracking enabled on RLBF models, they get much lower ASR than BSAFE+
>
> [1] https://artificialintelligenceact.eu/
>
> [2] https://cloud.google.com/vertex-ai/generative-ai/docs/multimodal/gemini-for-filtering-and-moderation
>
> ## **Coverage of Adaptive Attacks**
> We did indeed test GCG and Decoding Parameters Exploits (DPE) in the adaptive case. DPE by itself is adaptive to our case without any changes necessary (Please take a look at section 5.1 in [1] and the reviewer-author discussions for that paper in [1])
>
> Although GCG can also be argued to be adaptive, [1] proposes a version that also optimizes for getting the model not to generate reset tokens (backtracking tokens in our case). We followed the same procedure in their Section 5.1.
>
> We have only seen conflicting/negligible improvements for the attack capability of GCG with this addition (similar observations can be also in Table 2 of [1]):
>
> |Benchmark|BSAFE+|BSAFE+(AGCG)|RLBF|RLBF(AGCG)|
> |---|---|---|---|---|
> |AdvBench|6.6|7.0|4.7|4.7|
> |HEx-PHI|5.7|5.4|4.3|4.3|
>
> _Attack Success Rate (%) GCG and Adaptive GCG on RLBF and BSAFE+ (lower is better)_
>
> [1] https://openreview.net/forum?id=Bo62NeU6VF
>
> ## **End-to-End Efficiency Comparison**
> End-to-end efficiency is one of our contributions, and perhaps as exciting as the safety improvements we are getting with RLBF. Furthermore, we agree that we should give more motivation on the degree of efficiency we might expect from "backtrack by ``x`` tokens" compared to rewriting the part to be replaced or deleted. Also, we include the more fine-grained backtracking information directly by the number of tokens to backtrack for the SFT data, we also hypothesized that it should make the SFT stage more effective while being token efficient by design.
>
> We trained 5 new baseline models with the BSAFE version of backtracking (using the same training data), Gemma2 with 2B and 9B, and LlaMA3 with 1B, 3B and 8B sizes.
>
> |Method|Gemma2-2B|Gemma2-9B|LLaMA3-1B|LLaMA3-3B|LLaMA3-8B|
> |---|---|---|---|---|---|
> |IT|25|28|24|28|27|
> |RL|23|24|22|25|25|
> |BSAFE|18|17|18|20|22|
> |BSAFE+|14|15|14|17|16|
> |**RLBF**|**2**|**2**|**1**|**2**|**1**|
>
> _ASR (%) of various models and methods on LMSYS (lower is better)_
>
> In cases where the BSAFE, BSAFE+ and RLBF models backtracked, the whole backtracking part completed by 68.3 and 72.6 less tokens for BSAFE+ and RLBF, respectively, compared to BSAFE. However, due to the mismatch in performance (just as we hypothesized), this comparison might not be directly applicable. Therefore, we also looked at the what would have happened if the BSAFE+ and RLBF methods backtracked in the same style as BSAFE. For this, perhaps not surprisingly, we see the same trend by needing 74.8 and 79.1 fewer tokens for BSAFE+ and RLBF, respectively.
>
> These additional results portray the end-to-end efficiency gains by both in training effectiveness during SFT stage in terms of performance, and token efficiency during backtracking scenarios. A reduction of ~70-80 tokens might imply around ~2.5-3.5 sec for highly optimized chatbots such as GPT-4 or DeepSeek R1 [1]. For edge devices [2], this might mean ~5-10 sec. Real-time voice chat systems might have even more degradation in user experience as additional models run on the top of generated tokens.
>
> [1] https://artificialanalysis.ai/models/gpt-4/providers
>
> [2] https://arxiv.org/abs/2403.20041
>
> ---
> ## Further Clarifications
>
> >In line 174, how do you inject violation at a contextually coherent location? Can you elaborate more? How do you ensure the violating content can be semantically coherent after part 1?
>
> **Response:** This is mentioned in Section 4.1.2, however, here is a more detailed explanation:
>
> In order to train models to be robust to where the violations might occur, we randomly choose a point in the response to discard the rest of the generation. Then, we utilize the given prompt in Appendix A.1 to generate a contextually coherent harmful continuation. In our experience and qualitative observations, due to LLMs dominantly being trained on coherent text, completing these partial responses, albeit with harmful continuations, do not result in many mistakes. We also quickly tested the coherency with another LLM to see if there are even rare mistakes in coherency for 1000 generated harmful continuations, and there were no reported errors.
>
> >Can you elaborate more on the setting of middle filling attacks?
>
> **Response:** Thank you for this important clarification request and we appreciate the opportunity to elaborate.
>
> Middle Filling Attack Methodology: Middle filling attacks exploit the "shallow safety" vulnerability in current LLMs by strategically placing harmful content after initially benign context. The attack works as follows:
>
> 1. Benign Initialization: The prompt begins with legitimate, safe content that passes initial safety filters
> 2. Mid-Generation Injection: Harmful or policy-violating content is then introduced in the middle of what appears to be a coherent response
> 3. Exploitation of Temporal Safety Gaps: Standard safety mechanisms primarily monitor initial prompt tokens and often fail to detect violations that emerge during generation, particularly when they appear contextually plausible.
>
> We classify the attack as being successful if the model generated content which was unsafe. For the RLBF models, the attack is successful if:
> - unsafe tokens appear and,
> - RLBF fails to backtrack so that the final output is clean.
>
> Broader Connection: This attack vector is particularly relevant in the context of LLM agents processing documents, where adversarial instructions can be embedded within seemingly benign content. For example, a document containing instructions that get a model to ignore previous requests and to reveal private information about the user.
>
> ---
> ## Conclusion
> We have worked hard to clarify and include new experiments in our rebuttal. We kindly ask that you will reconsider your score based on our response. Feel free to let us know if you'd like us to address anything further. Thank you again for your thoughtful feedback!

---

> > ### Comment · Reviewer_wAMH · 2025-08-03
> > **Response to Rebuttal**
> >
> > Thank you for your newly-added experiments. I increase the my rating by one.
> >
> > Please make sure the incorporate the new experiments and results (adaptive attacks, End-to-End Efficiency Comparison, safety critics) into the paper, add the necessary experimental details in the main body of the paper.

---

### Official Review · Reviewer_MaoP · 2025-07-06

**Clarity:** 4
**Significance:** 3
**Originality:** 2
**Rating:** 5
**Confidence:** 3

**Summary:**

The paper proposes BSAFE+ and RLBF, which is an extension to BSAFE method that addresses some of the existing issues including redundant tokens and drop of performance.

The authors examined the limitations and conduct a investigation on how it can further improve the method. They showed the results on MT-MF benchmark.

**Questions:**

When redacting harmful context, how do you represent the number k parts?

It seems to me that llm might not be able to accurately predict the number of tokens, considering the step of tokenization. Would it affect the results?

Minor: Missing reference in the some text.

The illustration of figure 1 seems confusing, as it doesn't show the difference of two methods, which I thought it would.

**Ethical Concerns:**

["NO or VERY MINOR ethics concerns only"]

**Final Justification:**

Thanks for the response.

**Quality:**

3

**Strengths And Weaknesses:**

(Pros)
1. The paper is clear and well-written. It describes the motivation and ways to amend the baseline clearly.

2. The proposed fix to some of existing issues is aligned conceptually with the limitations of existing method.

3. The results shows promising improvement against the baseline.

I like the setting and execution of the methods.

---

> ### Author Rebuttal · Authors · 2025-07-31
>
> We thank the reviewer for their thoughtful review and questions. We were pleased to hear that it “shows promising improvement” and proposes "fix to some of existing issues".
>
> The reviewer did not list significant weaknesses of our paper, only clarification questions. In the following, we clarify each question in detail.
>
> ---
>
> ## Clarifications to Reviewer's Questions:
> >When redacting harmful context, how do you represent the number k parts?
>
> **Response:**
>
> Although we provided the core mechanism in Section 4.1.1, due to page limitations, here we give a more detailed explanation of how we represent and handle the backtrack counts and violation segments:
> - **Backtrack Count Representation:** The number of tokens to remove is represented through the `[BACKTRACK_BY_X]` token, where X is an integer directly embedded in the special token. For example, `[BACKTRACK_BY_5]` indicates that exactly 5 preceding tokens should be removed. This integer X is learned during SFT training where the model sees ground truth examples mapping violation contexts to their precise token counts.
> - **Multiple Violation Segments Handling:** When there are multiple harmful segments (k parts) within a single response, our approach handles them sequentially. The model can generate multiple backtrack signals throughout the generation process - each `[CATEGORY]` and `[BACKTRACK_BY_X]` pair addresses one violation segment at a time. The RL critic monitors the entire generation and can detect multiple violations, allowing the model to learn to backtrack from each harmful segment individually.
> - **Number Tokenization:** The actual integer in `[BACKTRACK_BY_X]` is tokenized as part of the special token vocabulary. During training, we ensure these tokens (e.g., `[BACKTRACK_BY_1]`, `[BACKTRACK_BY_2]`, etc.) are added to the model's vocabulary for common backtrack lengths. This eliminates any tokenization ambiguity since the entire backtrack command is a single, well-defined token.
> - **Violation Segment Boundaries:** Rather than pre-dividing harmful content into k predetermined parts, our approach dynamically identifies violation boundaries during generation. The critic determines the span of violating tokens, and the model learns to predict the appropriate backtrack count based on the detected violation's actual token length, providing more flexible and accurate correction than fixed segmentation approaches.
>
> These design choices collectively enable the model to precisely identify and efficiently correct harmful content segments without complex calculations or ambiguous representations, leading to the strong empirical performance demonstrated across our benchmarks.
>
> >It seems to me that llm might not be able to accurately predict the number of tokens, considering the step of tokenization. Would it affect the results?
>
> **Response:**
>
> This is an excellent question that touches on a fundamental aspect of our approach. However, tokenization complexity is not a significant issue for RLBF, and we address this concern through multiple mechanisms:
> - **SFT Stage Token Count Learning:** During the Supervised Fine-Tuning phase (Section 4.1.2), the model receives explicit supervision for accurate token counting. When we inject violating segments of length `|v|` into safe responses, the model learns to predict the exact backtrack count from ground truth examples. This creates a direct mapping between violation contexts and their corresponding token counts. Since the SFT data generation process knows the precise tokenization of both the violation and surrounding context, the model learns this mapping through thousands of training examples across diverse violation types and lengths.
> - **RL Stage Safeguards:** The RL stage provides additional safeguards against tokenization errors. As described in Section 4.2.2, our reward function specifically penalizes incoherent text with `R_final(τ) = -0.2` when the post-backtrack continuation is "NOT safe, OR is incoherent, OR fails to be useful." This negative reward signal teaches the model that incorrect token counting leading to incoherent text is undesirable. Additionally, we maintain an SFT mixture during RL training (`λ_SFT L_SFT_guidance(θ)` in the total loss function) to prevent the model from forgetting the precise token counting behavior learned during SFT.
> - **Inherent Token Access:** Critically, the model doesn't need to perform complex tokenization calculations. During generation, the model has direct access to the individual tokens in its input and generation history - the very tokens it needs to count for backtracking. The model operates at the token level, so determining "backtrack by X tokens" is simply a matter of referencing the existing token sequence, not performing external tokenization operations.
> - **Analogy to Standard Generation:** This challenge is analogous to the broader difficulty of coherent text generation in LLMs. Generating coherent, contextually appropriate text across long sequences is inherently complex, yet through training, LLMs achieve exceptional coherency and rarely produce generation artifacts. Similarly, learning to count tokens for backtracking is a learnable skill that becomes reliable through proper training signals.
> - **Qualitative Observations:** In our experiments, we qualitatively observe no significant issues with incorrect token counting leading to problematic backtracking. The consistently low attack success rates (1-7% across benchmarks in Tables 1-2) and preserved utility metrics (Table 3) suggest that any tokenization prediction errors are rare and do not meaningfully impact the system's effectiveness.
>
> The empirical success of RLBF demonstrates that token count prediction, while requiring careful training design, is not a practical limitation for this safety approach.
>
> >The illustration of figure 1 seems confusing, as it doesn't show the difference of two methods, which I thought it would.
>
> **Response:**
>
> Thank you for this insightful feedback about Figure 1's clarity. We appreciate the opportunity to clarify the illustration and acknowledge that the distinctions between the components could be more prominently highlighted.
> - **Current Figure Content:** The left side demonstrates our enhanced BSAFE+ data generation, where we inject violations into coherent, originally safe text and then show the backtracking mechanism. The right side illustrates the RL training phase where the critic provides real-time feedback on the model's live generation, identifying violations and guiding the learning of appropriate backtracking responses.
> - **Key Differences That Could Be More Prominent:** The figure contains the essential distinctions but could better highlight:
> 	- RLBF's streamlined "backtrack by X tokens" mechanism versus more complex repeat-and-edit approaches,
> 	- The dynamic critic feedback during RL training that operates on the model's own generation distribution rather than static training examples,
> 	- The integrated learning process where backtracking capability is refined through live critic evaluation rather than purely supervised learning.
> - **Methodological Innovations Present:** Figure 1 currently illustrates two critical components of our RLBF framework that represent significant advances over existing approaches. The BSAFE+ component shows our improved data generation strategy that preserves answer quality while providing precise supervision, and the RL component demonstrates how critic feedback enables in-distribution learning on the model's actual generation failures.
>
> Unfortunately, as per NeurIPS's rebuttal guidelines, we cannot share our updated figure. However, we have added visual cues or annotations to make the contrast more explicit, highlighting the simplified backtrack command format, emphasizing the real-time critic evaluation arrows, and adding a small comparison box showing the difference between static SFT training and dynamic RL refinement. These additions maintain the current information while making the methodological innovations more visually apparent. The core content demonstrating our contributions is present in the figure, but we agree that clearer visual emphasis on the distinctions would improve accessibility for readers seeking to understand the specific advances RLBF provides over prior work.
>
> ---
>
> ### Conclusion
> We are again thankful for your review and the suggested visual improvement points. If our clarifications and improvements helped, please consider updating your scores for our paper (e.g., Quality, Clarity, Significance and/or Originality).

---

### Note · Authors · 2025-08-16

Dear Area Chair and Reviewers,

We sincerely thank all reviewers for their constructive and thorough reviews of our work on Reinforcement Learning with Backtracking Feedback (RLBF). We are grateful for the time and effort each reviewer invested in understanding our contributions and providing valuable feedback that has strengthened our paper. We were particularly encouraged by the positive assessments noting our method's effectiveness in reducing attack success rates, maintaining model utility, and offering "potential applications in correcting mistakes in LLM generation beyond just safety."

In response to the thoughtful concerns raised during the review process, we conducted extensive additional experiments including critic quality evaluation across severity levels, comprehensive comparisons with Circuit Breakers across all benchmarks, end-to-end efficiency measurements demonstrating token savings, and validation of robustness against adaptive attacks. We are pleased that these additions, along with our detailed clarifications on implementation details, reward function design, and the backtracking mechanism, have addressed the reviewers' concerns. We appreciate that multiple reviewers acknowledged these improvements and updated their assessments accordingly. We will ensure all new experimental results and clarifications are properly incorporated into the camera-ready version, including moving key implementation details from supplementary to main text as suggested. Thank you again for your valuable insights that have helped us present RLBF's contributions more clearly and comprehensively.

Best regards,

The Authors

---

### Decision · Program_Chairs · 2025-09-17

**Decision:**

Accept (poster)

**Comment:**

This paper proposes Reinforcement Learning with Backtracking Feedback (RLBF), a framework to improve the safety of LLMs by learning to dynamically correct their own errors during RL. BSAFE+ is further proposed as a data creation technique by injecting violations into coherent, originally safe text, providing more effective initial training for the backtracking mechanism. The experiments demonstrate that RLBF significantly reduces attack success rates across diverse benchmarks and model scales.

Overall, the paper has the following strengths:
- The paper is well-written, with clear motivation and explanation of how the method improves existing baselines.
- RLBF introduces a targeted backtracking mechanism that removes only violating spans, making safety correction efficient.
- Strong empirical validation across attacks and model scales, showing reduced attack success rates while preserving standard benchmark performance.
- The backtracking approach is effective beyond safety (e.g., correcting mistakes in generation generally) and outperforms alternatives in reducing ASR while maintaining accuracy.

The paper initially has the following weaknesses:
- Several reviewers noted that important implementation details (e.g., data splits, training choices, critic evaluation) are lacking, making it difficult to assess reproducibility and the true source of performance gains.
- Multiple reviewers pointed out the vulnerability of depending on one LLM-based critic, whose quality is not sufficiently evaluated and may itself be adversarially exploitable or error-prone.
- Concerns were raised about incomplete experimental coverage, particularly the lack of tests against stronger adaptive attacks and limited baseline/contextual comparisons, which weakens claims of robustness and superiority.

After author rebuttal and discussions, most of the concerns have been addressed and the reviewers reached a consensus that the paper should be accepted. Therefore, the AC is happy to recommend acceptance.